# TRANSLATION BETWEEN WAVES, *wave2wave*

## ABSTRACT

The understanding of sensor data has been greatly improved by advanced deep learning methods with big data. However, available sensor data in the real world are still limited, which is called the opportunistic sensor problem. This paper proposes a new variant of neural machine translation *seq2seq* to deal with continuous signal waves by introducing the *window-based (inverse-) representation* to adaptively represent partial shapes of waves and the *iterative back-translation model* for high-dimensional data. Experimental results are shown for two real-life data: earthquake and activity translation. The performance improvements of one-dimensional data was about 46 % in test loss and that of high-dimensional data was about 1625 % in perplexity with regard to the original seq2seq.

## 1 INTRODUCTION

The problem of shortage of training data but can be supplied by other sensor data is called an *opportunistic sensor problem* (Roggen et al., 2013). For example in human activity logs, the video data can be missing in bathrooms by ethical reasons but can be supplied by environmental sensors which have less ethical problems. For this purpose we propose to extend the sequence-to-sequence (seq2seq) model (Cho et al., 2014; Sutskever et al., 2014; Dzmitry Bahdanau, 2014; Luong et al., 2015) to translate signal wave $x$ (*continuous time-series signals*) into other signal wave $y$. The straight-forward extension does not apply by two reasons: (1) the lengths of $x$ and $y$ are radically different, and (2) both $x$ and $y$ are high dimensionals.

First, while most of the conventional seq2seq models handle the input and output signals whose lengths are in the same order, we need to handle the output signals whose length are sometimes considerably different than the input signals. For example, the sampling rate of ground motion sensor is $100\mathbf{Hz}$ and the duration of an earthquake is about $10\mathbf{sec}$. That is, the length of the output signal wave is 10000 times longer in this case. Therefore, the segmentation along temporal axis and discarding uninformative signal waves are required. Second, signal waves could be high dimensionals; motion capture data has 129 dimensions and acceleormeter data has 18 dimensions. While most of the conventional seq2seq does not require the high-dimensional settings, meaning that it is not usual to translate multiple languages simultaneously, we need to translate signal waves in high dimensions into other signal waves in high dimensions simultaneously.

To overcome these two problems we propose 1) the window-based representation function and 2) the *wave2wave* iterative back-translation model in this paper. Our contributions are the following:

- We propose a sliding window-based seq2seq model *wave2wave* (Section 4.1),
- We propose the *wave2wave* iterative back-translation model (Section 4.2) which is the key to outperform for high-dimensional data.

## 2 RELATED WORKS

Related works include various encoder-decoder architectures and generative adversarial networks (GANs). First, the encoder-decoder architecture has several variations: (1) CNNs in both sides (Badrinarayanan et al., 2015), (2) RNNs in both sides (Cho et al., 2014; Sutskever et al., 2014; Dzmitry Bahdanau, 2014; Luong et al., 2015), or (3) one side is CNN and the other is RNN (Xu et al., 2015). When one side is related to autoregressive model (van den Oord et al., 2016), further variations are appeared. These architectures are considered to be distinctive. The pros of CNN is an

efficient extraction of features and overall execution while the pros of RNN is its excellent handling of time-series or sequential data. CNN is relatively weak in handling time-series data. In this reason, the time domain is often handled by RNN. The encoder-decoder architecture using CNNs in both sides is used for semantic segmentation (Badrinarayanan et al., 2015), image denoising (Mao et al., 2016), and super-resolution (Chen et al., 2018), which are often not related to time-series. In the context of time-series, GluonTS (Alexandrov et al., 2019) uses the encoder-decoder approach which aims at time-series prediction task where parameters in encoder and decoder are shared. Apart from the difference of tasks, our approach does not share the parameters in encoder and those in decoder. All the more our model assumes that the time-series multi-modal data are related to the multi-view of the same targeted object which results in multiple modalities. Second, among various GAN architectures, several GANs aims at handling time-series aspect. Vid2vid (Wang et al., 2018) is an extension of pix2pix (Isola et al., 2016) which aims at handling video signals. ForGAN (Koochali et al., 2019) aims at time-series prediction task.

## 3 SEQ2SEQ

**Architecture with context vector**   Let $x_{1:S}$ denotes a source sentence consisting of time-series $S$ words, i.e., $x_{1:S} = (x_1, x_2, \ldots, x_S)$. Meanwhile, $y_{1:T} = (y_1, \ldots, y_T)$ denotes a target sentence corresponding to $x_{1:S}$. With the assumption of a Markov property, the conditional probability $p(y_{1:T}|x_{1:S})$, translation from a source sentence to a target sentence, is decomposed into a time-step translation $p(y|x)$ as in (1):

$$\log p(y_{1:T}|x_{1:S}) = \sum_{t=1}^{T} \log p(y_t|y_{<t}, \mathbf{c}_t) \tag{1}$$

where $y_{<s} = (y_1, y_2, \ldots, y_{s-1})$ and $\mathbf{c}_s$ is a context vector representing the information of source sentence $x_{1:S}$ to generate an output word $y_t$.

To realize such time-step translation, the seq2seq architecture consists of (a) a RNN (Reccurent Neural Network) encoder and (b) a RNN decoder. The RNN encoder computes the current hidden state $\mathbf{h}_s^{\text{enc}}$ given the previous hidden state $\mathbf{h}_{s-1}^{\text{enc}}$ and the current input $x_s$, as in (2):

$$\mathbf{h}_s^{\text{enc}} = \text{RNN}_{\text{enc}}(x_s, \mathbf{h}_{s-1}^{\text{enc}}) \tag{2}$$

where $\text{RNN}_{\text{enc}}$ denotes a multi-layered RNN unit.

The RNN decoder computes a current hidden state $\mathbf{h}_t^{\text{dec}}$ given the previous hidden state and then compute an output $y_t$.

$$\mathbf{h}_t^{\text{dec}} = \text{RNN}_{\text{dec}}(\mathbf{h}_{t-1}^{\text{dec}}) \tag{3}$$

$$p_\theta(y_t|y_{<t}, \mathbf{c}_t) = \text{softmax}\Big(g_\theta(\mathbf{h}_t^{\text{dec}}, \mathbf{c}_t)\Big) \tag{4}$$

where $\text{RNN}_{\text{dec}}$ denotes a conditional RNN unit, $g_\theta(\cdot)$ is the output function to convert $\mathbf{h}_t^{\text{dec}}$ and $\mathbf{c}_t$ to the logit of $y_t$, and $\theta$ denotes parameters in RNN units.

With training data $\mathcal{D} = \{y_{1:T}^n, x_{1:S}^n\}_{n=1}^N$, the parameters $\theta$ are optimized so as to minimize the loss function $\mathcal{L}(\theta)$ of log-likelihood:

$$\mathcal{L}(\theta) = -\frac{1}{N} \sum_{n=1}^{N} \sum_{t=1}^{T} \log p_\theta(y_t^n|y_{<t}^n, \mathbf{c}_t) \tag{5}$$

or squared error:

$$\mathcal{L}(\theta) = \frac{1}{N} \sum_{n=1}^{N} \sum_{t=1}^{T} \Big(y_t^n - g_\theta(\mathbf{h}^{\text{dec}^n}_t, \mathbf{c}_t^n)\Big)^2 \tag{6}$$

**Global Attention**   To obtain the context vector $\mathbf{c}_s$, we use global attention mechanism (Luong et al., 2015). The global attention considers an alignment mapping in a global manner, between encoder hidden states $\mathbf{h}_s^{\text{enc}}$ and a decoder hidden step $\mathbf{h}_t^{\text{dec}}$.

$$a_t(s) = \text{align}(\mathbf{h}_t^{\text{dec}}, \mathbf{h}_s^{\text{enc}}) \tag{7}$$

$$= \frac{\exp(\text{score}(\mathbf{h}_t^{\text{dec}}, \mathbf{h}_s^{\text{enc}})}{\sum_s^T \exp(\text{score}(\mathbf{h}_t^{\text{dec}}, \mathbf{h}_s^{\text{enc}}))} \tag{8}$$

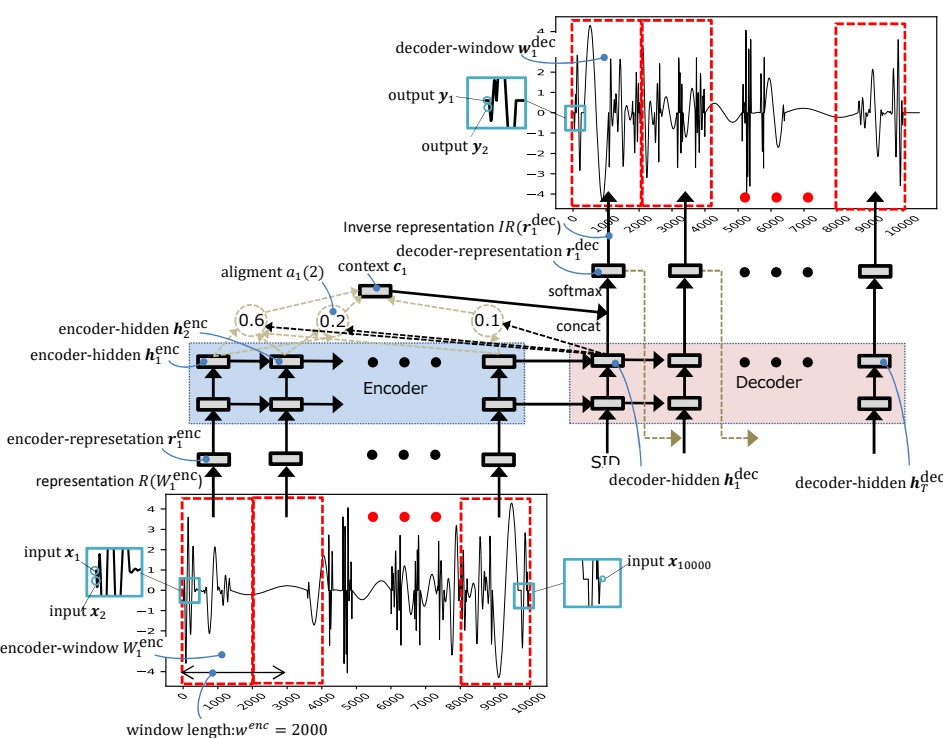

Figure 1: Overall architecture of our method, wave2wave, consisting RNN encoder and decoder with context vector and sliding window representation. Input and output time-series data are toy examples where the input is generated by combining sine waves with random magnitudes and periods. The output is the version of the input flipped horizontally.

where the score is computed by weighted inner product as follows

$$\text{score}(\mathbf{h}_t^{\text{dec}}, \mathbf{h}_s^{\text{enc}}) = \mathbf{h}_t^{\text{dec}\top} W_a \mathbf{h}_s^{\text{enc}} \tag{9}$$

where the weight parameter $W_a$ is obtained so as to minimize the loss function $\mathcal{L}(\theta)$. Then, the context vector $\mathbf{c}_t$ is obtained as a weighted average of encoder hidden states as

$$\mathbf{c}_t = \sum_{s=1}^{S} a_t(s) \mathbf{h}_s^{\text{enc}} \tag{10}$$

## 4 PROPOSED METHOD: WAVE2WAVE

The problems of global attention model are that (1) the lengths of input and output are radically different, and that (2) both input and output sequences are high dimensionals. For example in activity translation, there are 48 motion sensors and 3 accelerometer sensors. Their frequency rates are as high as 50Hz and 30Hz respectively. Therefore, the number of steps $S$, $T$ in both encoder and decoders are prohibitively large so that the capturing information of source sentence $x_{1:S}$ is precluded in the context vector $\mathbf{c}$.

### 4.1 WINDOW-BASED REPRESENTATION

Let us consider the case that source and target sentences are multi-dimensional continuous time-series, signal waves, as shown in Figure 1 [1]. That is, each signal at time-step $x_{1:S}$ is expressed as $d_\mathbf{x}$-dimensional vector $\mathbf{x}_s$—there are $d_\mathbf{x}$ sensors in the source side. Then a source signal wave $\mathbf{x}_{1:S}$ consists of $S$-step $d_\mathbf{x}$-dimensional signal vectors, i.e., $\mathbf{x}_{1:S} = (\mathbf{x}_1, \mathbf{x}_2, \dots, \mathbf{x}_S)$.

---

[1]We note that signal waves in Figure 1 are depicted as one-dimensional waves for clear visualization.

To capture an important shape informaion from complex signal waves (see Figure 1), we introduce trainable window-based representation function $R(\cdot)$ as

$$\mathbf{r}_{s'}^{\mathrm{enc}} = R(W_{s'}^{\mathrm{enc}}) \tag{11}$$

where $W_{s'}^{\mathrm{enc}}$ is a $s'$-th window with fixed window-width $w^{\mathrm{enc}}$, expressed as $d_{\mathbf{x}} \times w^{\mathrm{enc}}$-matrix as

$$W_{s'}^{\mathrm{enc}} = \left[\mathbf{x}_{w^{\mathrm{enc}}(s'-1)+1}, \mathbf{x}_{w^{\mathrm{enc}}(s'-1)+2}, \ldots, \mathbf{x}_{w^{\mathrm{enc}}(s'-1)+w^{\mathrm{enc}}}\right], \tag{12}$$

and $\mathbf{r}_{s'}^{\mathrm{enc}}$ is extracted representation vector inputted to the seq2seq encoder as shown in Figure 1 —the dimension of $\mathbf{r}^{\mathrm{enc}}$ is the same as the one of the hidden vector $\mathbf{h}^{\mathrm{enc}}$.

Similarly, to approximate the complex target waves well, we introduce inverse representation function, $R^{-1}(\cdot)$ which is separately trained from $R^{-1}(\cdot)$ as

$$W_{t'}^{\mathrm{dec}} = R^{-1}(\mathbf{r}_{t'}^{\mathrm{dec}}) \tag{13}$$

where $\mathbf{r}_{t'}^{\mathrm{dec}}$ is the $t'$-th output vector from seq2seq decoder as shown in Figure 1, and $W_{t'}^{\mathrm{dec}}$ is a window matrix which is corresponding to a partial wave of target waves $\mathbf{y}_{1:T} = (\mathbf{y}_1, \ldots, \mathbf{y}_T)$.

The advantage of window-based architecture are three-fold: firstly, the number of steps in both encoder and decoder could be largely reduced and make the seq2seq with context vector work stably. Secondly, the complexity and variation in the shape inside windows are also largely reduced in comparison with the entire waves. Thus, important information could be extracted from source waves and the output sequence could be accurately approximated by relatively simple representation $R(\cdot)$ and inverse-representation $R^{-1}(\cdot)$ functions respectively. Thirdly, both representation $R(\cdot)$ and inverse-representation $R^{-1}(\cdot)$ functions are trained end-to-end manner by minimizing the loss $\mathcal{L}(\theta)$ where both functions are modeled by fully-connected (FC) networks.

Figure 1 depicts the overall architecture of our wave2wave with an example of toy-data. The wave2wave consists of encoder and decoder with long-short term memory (LSTM) nodes in their inside, representation function $R(W_{s'}^{\mathrm{enc}})$ and inverse-representation function $R^{-1}(W_{t'}^{\mathrm{dec}})$. In this figure, one-dimensional 10000-time-step continuous time-series are considered as an input and an output and the width of window is set to 2000— there are 5 window steps for both encoder and decoder, i.e., $w^{\mathrm{enc}} = w^{\mathrm{dec}} = 2000$ and $S' = T' = 5$. Then, $1 \times 2000$ encoder-window-matrix $W_{s'}^{\mathrm{enc}}$ is converted to $d_{\mathbf{r}}$ dimensional encoder-representation vector $\mathbf{r}_{s'}^{\mathrm{enc}}$ by the representation function $R(W_{s'}^{\mathrm{enc}})$. Meanwhile, the output decoder, $d_{\mathbf{r}}$ dimensional decoder-representation $\mathbf{r}_{t'}^{\mathrm{dec}}$, is converted to $1 \times 2000$ decoder-window-matrix $W_{t'}^{\mathrm{dec}}$ by the inverse representation function $R^{-1}(\mathbf{r}_{t'}^{\mathrm{dec}})$.

## 4.2 WAVE2WAVE ITERATIVE MODEL

We consider two different ways to tackle with high-dimensional sensor data. Since NMT for machine translation handles embeddings of words, the straightforward extention to high-dimensional settings uses the $d_{\mathbf{x}}$-dimensional source signal at the same time step as source embeddings, and the $d_{\mathbf{y}}$-dimensional target signal at the same time step as target embeddings. We call this a wave2wave model, i.e. our standard model. Alternatively, we can build $d_{\mathbf{y}}$ independent embeddings separately for corresponding individual 1-dimensional target signal at each time step while we use the same $d_{\mathbf{x}}$-dimensional source signal embeddings. We call this a Wave2WaveIterative model. We suppose that the former model would be effective when sensor data are correlated while the latter model would be effective when sensor data are independent. Algorithm 1 shows the latter algorithm.

---

**Algorithm 1:** Wave2waveIterative model

---

**Data:** $\mathrm{src}_{d_{\mathbf{x}} \times S}$, $\mathrm{tgt}_{d_{\mathbf{y}} \times T}$, $e_{src} \leftarrow \mathbf{x}^{d_{\mathbf{x}}}$, $e_{tgt_j} \leftarrow \mathbf{y}_j^{d_{\mathbf{y}}}$

**def** *trainWave2WaveIterative($e_{src} \times S$, $e_{tgt} \times T$)***:**

    **for** $j = (1, d_{\mathbf{y}})$ **do**

        f(j) = trainWave2Wave($e_{src} \times S$, $e_{tgt_j} \times T$);

    **end**

---

Note that the embedding $e_{src}$ is equivalent to $d_{\mathbf{x}}$-dimensionally decomposed representation of $\mathbf{r}_{s'}^{\mathrm{enc}}$, and $e_{tgt}$ is equivalent to $d_{\mathbf{y}}$-dimensionally decomposed representation of $\mathbf{r}_{t'}^{\mathrm{dec}}$. The back-translation is a technique to improve the performance by bi-directional translation removing the noise under a *neutral-biased* translation (Hoang et al., 2018). We deploy this technique which we call the wave2wave iterative back-translation model.

| method | train loss | test loss |
|---|---|---|
| simple encoder-decoder $d_{\mathbf{z}} = 100$ | 1.13 | 0.53 |
| simple encoder-decoder $d_{\mathbf{z}} = 500$ | 0.90 | 0.47 |
| simple encoder-decoder $d_{\mathbf{z}} = 1000$ | 0.41 | 0.63 |
| simple seq2seq $w^{\mathrm{enc}} = W^{\mathrm{dec}} = 500$ | 9.27 | 2.87 |
| simple seq2seq $w^{\mathrm{enc}} = W^{\mathrm{dec}} = 1000$ | 9.87 | 2.79 |
| simple seq2seq $w^{\mathrm{enc}} = W^{\mathrm{dec}} = 2000$ | 6.82 | 2.60 |
| wave2wave $w^{\mathrm{enc}} = W^{\mathrm{dec}} = 500$ | 0.67 | 0.44 |
| wave2wave $w^{\mathrm{enc}} = W^{\mathrm{dec}} = 1000$ | 0.17 | 0.34 |
| wave2wave $w^{\mathrm{enc}} = W^{\mathrm{dec}} = 2000$ | 0.25 | 0.43 |

Table 1: Mean squared loss of simple encoder-decoder methods, simple seq2seq methods and our wave2wave in earthquake ground motion data

## 5 EVALUATION ON REAL-LIFE DATA: GROUND MOTION TRANSLATION

In this section, we apply our proposed method, wave2wave, to predict a broadband-ground motion from only its long-period motion, caused by the same earthquake. In this section, wave2wave translates one dimensional signal wave into one dimensional signal wave.

Ground motions of earthquakes cause fatal damages on buildings and infrastructures. Physics-based numerical simulators are used to generate ground motions at a specific place, given the property of earthquake, e.g., location and scale to estimate the damages on buildings and infrastructures (Iwaki & Fujiwara, 2013). However, the motion generated by simulators are limited only long periods, longer than 1 second due to heavy computational costs, and the lack of detailed knowledge of the subsurface structure.

A large amount of ground motion data have been collected by K(kyosin)-NET over the past 20 years in Japan. Machine learning approaches would be effective to predict broadband-ground motions including periods less than 1 second, from simulated long period motions. From this perspective, we apply our method wave2wave to this problem by setting long-ground motion as an input and broadband-ground motion as an output, with the squared loss function $\mathcal{L}(\theta)$.

As for training data, we use 365 ground motion data collected at the observation station, IBR011, located at Ibaraki prefecture, Japan from Jan. 1, 2000 to Dec. 31, 2017—originally there are 374 data but 10 data related Tohoku earthquakes and the source deeper than 300m are removed. As for testing, we use 9 ground-motion data of earthquakes occurred at the beginning of 2018.

In addition, both long and broadband ground motion data are cropped to the fixed length, i.e., $s = t = 10000$ms and its amplitude is smoothed using RMS (Root Mean Square) envelope with 200ms windows to capture essential property of earthquake motion. Moreover, as for data augmentation, in-phase and quadrature components, and those absolute values are extracted from each ground motion. That is, there are totally $365 \times 3$ training data. Figure 2a shows an example of 3 components of a ground motion of earthquake occurred on May 17, 2018, Chiba in Japan, and corresponding RMS envelopes.

Table 1 depicts the mean-squared loss of training of three methods, simple encoder-decoder, simple seq2seq, and our proposed method with the same setting as the toy data except $\mathbf{h}^{\mathrm{enc}} = \mathbf{h}^{\mathrm{dec}} = 50$. This table shows that our wave2wave methods basically outperform other methods although wave2wave with the small window-width $w^{\mathrm{enc}} = w^{\mathrm{dec}} = 500$ is lost by simple encoder-decoder with large hidden layer $d_{\mathbf{z}} = 1000$ in train loss. This indicates that window-based representation and inverse-representation functions are helpful similarly in toy data.

Figure 2b depicts examples of predicted broadband ground motions of earthquakes occurred on Jan. 24 and May 17, 2018. These show that our method wave2wave predict enveloped broadband ground motion well given long-period ground motion although there is little overfitting due to small training data.

It is expected that predicted broadband-motion combined with simulated long-period motion could be used for more accurate estimation of the damages on buildings and infrastructures.

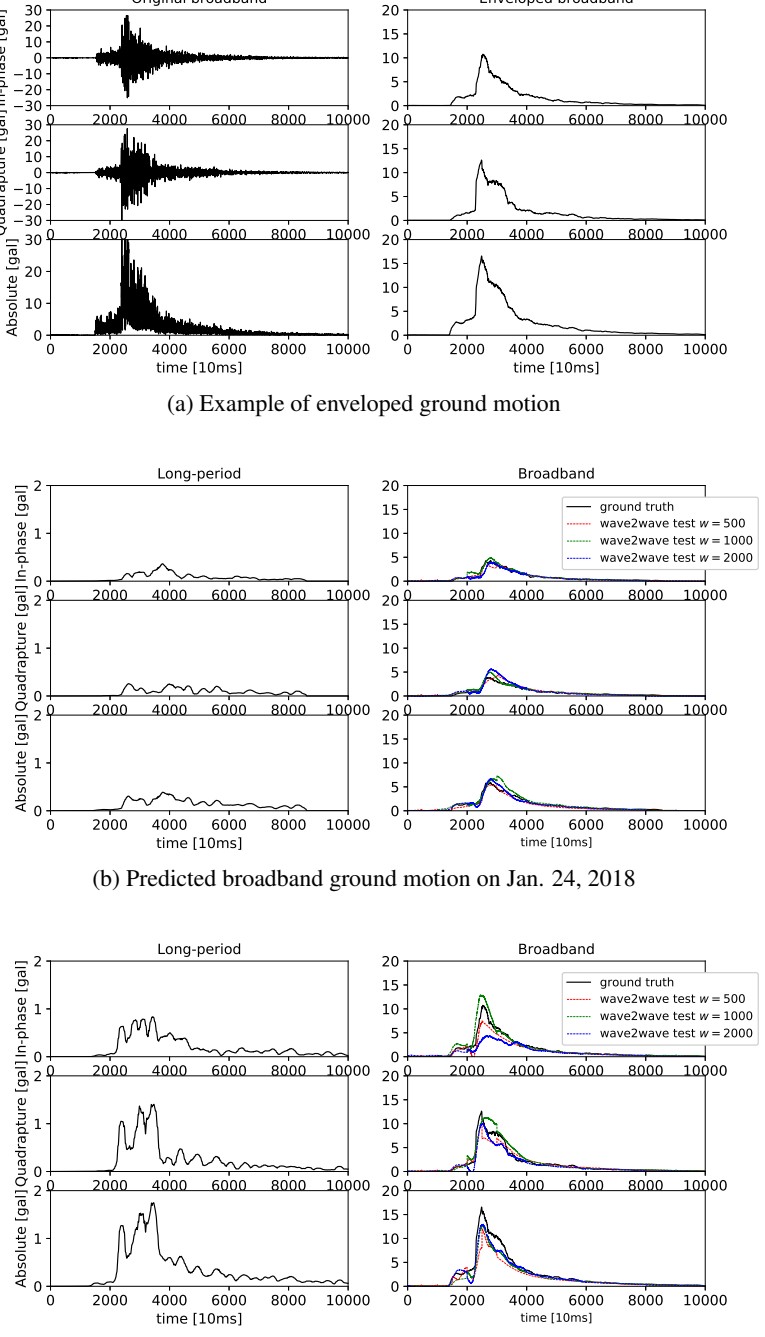

(a) Example of enveloped ground motion

(b) Predicted broadband ground motion on Jan. 24, 2018

(c) Predicted broadband ground motion on May. 17, 2018

Figure 2: *top*: Example of original and enveloped ground motion data with in-phase, quadrature components and these absolute values. *middle and bottom*: predicted broadband ground motion by our methods wave2wave for earthquakes occurred on Jan. 24, 2018 and May. 17, 2018.

# 6    EVALUATION ON REAL-LIFE DATA: ACTIVITY TRANSLATION

This section deploys wave2wave for activity translation (Refer Figure 3). Until the previous section, the signals were one dimensions. The signals in this section are high-dimensional in their inputs

as well as outputs. The dimensions of motion capture, video, and accelerometer are 129, 48, and 18 dimensions, respectively, in the case of MHAD dataset[2]. Under the mild assumption that the

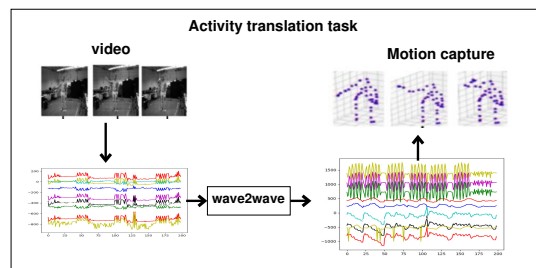

Figure 3: Figure shows activity translation task and activity recognition task which we conduct experiments.

targeted person which are recorded in three different modalities, including motion capture, video, and accelerometer, are synchronized and the noise such as the effect of other surrounding persons is eliminated. Hence, we assume that each signal shows one of the multi-view projections of a single person. That is, we can intuitively think that they are equivalent. Under this condition, we do a translation from motion capture to video (Similarly, accelerometer to motion capture, and video to accelerometer, and these inverse directions).

### 6.0.1 OVERALL ARCHITECTURE

**Wave Signal** Figure 3 shows that motion capture and video can be considered as wave signal. When video is input, $W_{s'}^{\text{enc}}$ takes the form of pose vectors which are converted by OpenPose library (Cao et al., 2017). Then, this representation is convereted into the window representation by $\text{R}(W_{s'}^{\text{enc}})$. When motion capture is input, $W_{s'}^{\text{enc}}$ takes the form of motion capture vectors. In this way we used these signals for input as well as output for wave2wave. The raw output are reconstructed by $\text{R}^{-1}(W_{t'}^{\text{dec}})$ for the output of representation $W_{t'}^{\text{dec}}$.

**Wave Signal Dimensionality Reduction** As an alternative to use FC layer before the input, we use the clustering algorithm, specifically an affinity propagation (Frey & Dueck, 2007), in order to reduce the size of representation as a whole. While most clustering algorithms need to supply the number of clusters beforehand, this affinity propagation algorithm solves the appropriate cluster number as a result.

**Multi-Resolution Spatial Pyramid** Additinaly structures in wave2wave is the multi-scalability since the frame rate of multimodal data are considerably different. We adopted the approach of multi-resolution spatial pyramid by a dynamic pose (Neverova et al., 2014). We assume that the sequence of frames across modalities is synchronized and sampled at a given temporal step $v$ and concatenated to form a spatio-temporal 3-d volume.

### 6.0.2 EXPERIMENTAL EVALUATION

**Experimental Setups** We used the MHAD dataset from Berkeley. We used video, accelerometer, and mocap modalities. We used video with Cluster-01/Cam01-02 subsets, and the whole mocap (optical) and accelerometer data with 12 persons/5 trials. Video input was preprocessed by Open-Pose which identifies 48 dimensions of vectors. Optical mocap had the position of the keypoints whose dimension was 129. Accelerometer were placed in 6 places in the body whose dimension was 18. We used the parameters in wave2wave with cross entropy loss function $\mathcal{L}(\theta) = -\frac{1}{N}\sum_{n=1}^{N}$ $\log p_\theta(y_{1:T}^n|x_{1:S}^n)$ with LSTM modules 500, embedding size 500, dropout 3, maximum sentence length 400, and batch size 100. We used Adam optimizer. We used $v = 2, 3, 4$ for multi-resolution spatial pyramid. We used the same parameter set for wave2wave interactive model. We use Titan Xp.

---

[2]http://tele-immersion.citris-uc.org/berkeley_mhad.

| $(w^{\mathrm{enc}}, W^{\mathrm{dec}})$ | ppl $d_{\mathbf{z}} = 1$ | ppl $d_{\mathbf{z}} = 129$ | ppl $d_{\mathbf{z}} = 1$ | ppl $d_{\mathbf{z}} = 129$ | ppl $d_{\mathbf{z}} = 1$ | ppl $d_{\mathbf{z}} = 129$ |
|---|---|---|---|---|---|---|
| | seq2seq baseline | | seq2seq clustering | | | |
| | 58000.42 | 52000.33 | 5.20 | 30.22 | | |
| | wave2wave | | wave2waveIte | | wave2waveIteBacktrans | |
| (1,16) | 2.13 | 19.74 | 2.13 | 4.72 | 2.13 | 4.73 |
| (5,80) | 0.33 | 10.73 | 0.33 | 3.44 | 0.32 | 3.40 |
| (10,160) | 0.42 | 11.28 | 0.42 | 3.49 | 0.42 | 3.48 |
| (20,320) | 0.72 | 13.67 | 0.72 | 3.78 | 0.72 | 3.75 |
| (30,480) | 1.21 | 15.03 | 1.21 | 4.11 | 1.21 | 4.11 |
| (60,960) | 4.30 | 35.98 | 4.30 | 6.81 | 4.30 | 6.82 |

Table 2: Figure shows major experimental results for acc2moc.

**Human Understandability**   One characteristic of activity translation can be observed in the direction of wave2wave translation with accelerometer to video, e.g. acc2cam. That is, the accelerometer data takes the form that is not understandable by human beings by its nature but translation to video makes this visible. By selecting 50 test cases, the human could understand 48 cases. 96 % is fairly good. The second characteristic of activity translation is opportunistic sensor problem, e.g. when we cannot use video camera in bathrooms, we use other sensor modality, e.g. accelerometer, and then translate it to video which can use at this *opportunity*. This corresponds to the case of acceleromter to video, e.g. acc2cam. We conduct this experiments. Upon watching the video signals on a screen we could observe the basic human movements. By selecting 50 test cases, the human could understand 43 cases.

**Experimental Results**   Major experimental results are shown in Table 2. We used $w^{\mathrm{enc}} = \{1, 5, 10, 20, 30, 60\}$. For each window size we measured one target with perplexity (ppl) and the whole target with perplexity (ppl). We compared several wave2wave results with (1) the seq2seq model without dimensionality reduction (via clustering), (2) the seq2seq model with dimensionality reduction. All the experiments are done with the direction from accelerometer to motion capture (acc2moc).

Firstly, the original seq2seq model did not work well without dimensionality reduction of input space. The perplexity was 58000.42. This figure suggests that the optimization of deep learning did not go progress due to the complexity of the training data or the bad initialization. However, the results were improved fairly well if we do dimensionality reduction using clustering. This figure is close to the results by wave2wave (iterative) with $w^{\mathrm{enc}} = 60$.

Secondly, $w^{\mathrm{enc}} = 5$ performed better than other window size for perplexity when $d_{\mathbf{z}} = 1$. When this became high dimensional, the wave2wave iterative model performed better than the wave2wave mode: 3.44 vs 10.73 in perplexity. Since motion capture has $d_{\mathbf{z}} = 129$ dimensions, the representation space becomes $R^{d_{\mathbf{z}}}$ when we let $R$ denote the parameter space of one point in motion capture. Compared with this the wave2wave iterative model equipped with the representation space linear with $R$. The wave2wave iterative model has an advantage in this point. Moreover, the wave2wave iterative back-translation model made the best score in perplexity when $d_{\mathbf{z}} = 1$ as well as $d_{\mathbf{z}} = 129$.

## 7   CONCLUSION

We proposed a method to translate between waves *wave2wave* with a sliding window-based mechanism and iterative back-translation model for high-dimensional data. Experimental results for two real-life data show that this is positive. Performance improvements were about 46 % in test loss for one dimensional case and about 1625 % in perplexity for high-dimensional case using the iterative back-translation model.

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
