# OpenReview forum: "Translation Between Waves,  wave2wave"
_ICLR.cc/2020/Conference — Reject_

### Official Review · AnonReviewer3 · 2019-10-20
**Official Blind Review #3**

**Rating:** 1

**Review:**

The authors propose a variant of sequence to sequence models that operates on the fixed segments (waves). The paper contains significant flaws both in its writing and its experimental setup.

1) Authors write: "For example in human activity logs, the video data can be missing in bathrooms by ethical reasons but can be supplied by environmental sensors which have less ethical problems.". I believe that this statement is completely irrelevant to the paper.

2) Authors write: "The straight-forward extension does not apply by two reasons: (1) the lengths of x and y are radically different, and (2) both x and y are high dimensionals." This statement is not justified and to the contrary of authors statement there has been recent work on wave2wave modeling including end-to-end speech-to-speech neural translation by Jia et al 2019, Liu et al 2019.

3) Authors write: "CNN is relatively weak in handling time-series data." Again this statement is not justified and to the contrary there has been work by Gehring et al 2017 showing success of CNNs in neural machine translation and Bai et al 2018 showing success of Temporal Convolutional Nets on various sequence modeling tasks.

4) Several typos in the paper: Reccurent -> Recurrent on page 2, informaion -> information on page 5
I believe that this paper is far from worthy of being considered to be accepted by ICLR.

5) Authors claim that they learn several inverse function R^{-1} to capture target waves. On the first glance it may seem that they invert the same weight matrix used for learning source waves R, however upon closer inspection it seems that they learn separate weight matrices for source and target weights which makes inverse notation very confusing.

6) I believe experimental setup is flawed. Authors consider the tasks that are not used by the community which makes it impossible to compare their method to related literature. Furthermore it is not clear to me how capturing the sequence semi-autoregressively (by modeling chunks of waves) produces such as big boost over vanilla sequence to sequence baseline.

**Experience Assessment:**

I have published in this field for several years.

**Review Assessment: Checking Correctness Of Derivations And Theory:**

I assessed the sensibility of the derivations and theory.

**Review Assessment: Checking Correctness Of Experiments:**

I assessed the sensibility of the experiments.

**Review Assessment: Thoroughness In Paper Reading:**

I read the paper at least twice and used my best judgement in assessing the paper.

---

### Official Review · AnonReviewer2 · 2019-10-22
**Official Blind Review #2**

**Rating:** 1

**Review:**

In this paper, the authors propose modifications to baseline seq-to-seq systems for wave-to-wave translation. To handle possibly long inputs and outputs, as well as significant length differences, they propose to use sliding windows. For high-dimensional outputs, they use an iterative approach predicting each dimension independently. They evaluate their models on earthquake data on on activity translation (video to motion capture).

I would reject this paper because it largely ignores the litterature on speech recognition, text-to-speech synthesis and speech-to-speech translation (e.g. Jia et al. Direct speech-to-speech translation with a sequence-to-sequence model). I would suggest to the authors to familiarize themselves with work presented at conferences such as Interspeech and ICASSP.

The contributions of the paper are unfortunately minimal. Window-based representations are already used, for example to obtain spectrograms. Predicting each feature independently appears very inefficient and mostly suitable for small datasets.

While it wouldn't change my rating of the paper, there are many typos (e.g. acceleormeter, informaion) that could easily be corrected.

**Experience Assessment:**

I have read many papers in this area.

**Review Assessment: Checking Correctness Of Derivations And Theory:**

I assessed the sensibility of the derivations and theory.

**Review Assessment: Checking Correctness Of Experiments:**

I assessed the sensibility of the experiments.

**Review Assessment: Thoroughness In Paper Reading:**

I made a quick assessment of this paper.

---

### Official Review · AnonReviewer1 · 2019-10-23
**Official Blind Review #1**

**Rating:** 3

**Review:**

This work explains how to use a seq2seq encoder-decoder neural network on the case of multivariate time series. The authors name this particular application of seq2seq the wave2wave network. Given a multivariate time series covering a time interval, it is split into subintervals of equal length, such that each block is a matrix. This matrix becomes an input into a recurrent encoder. On the decoder side, the similar matrix is produced at the output. The proposed neural network is tested on two data sets: an earthquake and activity translation.
Strenghts:
+ application on activity translation and evaluation by human raters is interesting
+ the proposed work is a nice application of an encoder-decoder architecture in case of multivariate time series
Weaknesses:
- there is no methodological novelty. The proposed network uses a standard architecture and the authors use straightforward partitioning of the signal in equal-sized windows. Due to this, inventing a new name wave2wave is not fully justified.
- the paper has many grammar mistakes and typos (for example, first sentence in the introduction seems to miss a few words). The writing is vague and imprecise and occasionally interferes with understanding. It would be important for the authors to carefully proofread the paper for language and clarity of presentation.
- the experimental results are not explained in a sufficient detail. I did not understand what is the objective of the earthquake application and how does the data set looks like (e.g., what do you mean by “365 x 3 training data”). The details of experimental design, architecture and hyperparameter choice are not provided in the earthquake application. It is not clear what the authors mean by “simple encoder-decoder” and “simple seq2seq”, I guess those are some baselines. For the activity application, it would be useful to see some illustration of the neural network output when video is predicted from motion capture. It would help to clarify the accuracy measures – perplexity and human understability. How many humans were involved in rating and what were they asked to do? How did the humans rate the non-video outputs?

Overall, given the quality of writing and lack of methodological novelty this paper is not ready for publication.

**Experience Assessment:**

I have published one or two papers in this area.

**Review Assessment: Checking Correctness Of Derivations And Theory:**

I assessed the sensibility of the derivations and theory.

**Review Assessment: Checking Correctness Of Experiments:**

I assessed the sensibility of the experiments.

**Review Assessment: Thoroughness In Paper Reading:**

I read the paper at least twice and used my best judgement in assessing the paper.

---

### Decision · Program_Chairs · 2019-12-19

**Decision:**

Reject

**Comment:**

The paper considers the task of sequence to sequence modelling with multivariate, real-valued time series.
The authors propose an encoder-decoder based architecture that operates on fixed windows of the original signals.

The reviewers unanimously criticise the lack of novelty in this paper and the lack of comparison to existing baselines.
While Rev #1 positively highlights human evaluation contained in the experiments, they nevertheless do not think this paper is good enough for publication as is.
The authors did not submit a rebuttal.

I therefore recommend to reject the paper.